# Low-Frequency Raman Spectroscopy: An Exceptional Tool for Exploring Metastability Driven States Induced by Dehydration

**DOI:** 10.3390/pharmaceutics15071955

**Published:** 2023-07-15

**Authors:** Yannick Guinet, Laurent Paccou, Alain Hédoux

**Affiliations:** UMR 8207-UMET-Unité Matériaux et Transformations, Univ. Lille, CNRS, INRAE, Centrale Lille, F-59000 Lille, France; yannick.guinet@univ-lille.fr (Y.G.); laurent.paccou@univ-lille.fr (L.P.)

**Keywords:** theophylline, caffeine, driven state, low-frequency Raman spectroscopy, dehydration kinetics

## Abstract

The use of low-frequency Raman spectroscopy (LFRS; ω < 150 cm^−1^) is booming in the pharmaceutical industry. Specific processing of spectra is required to use the wealth of information contained in this spectral region. Spectra processing and the use of LFRS for analyzing phase transformations in molecular materials are detailed herein from investigations on the devitrification of ibuprofen. LFRS was used to analyze the dehydration mechanism of two hydrates (theophylline and caffeine) of the xanthine family. Two mechanisms of solid-state transformation in theophylline were determined depending on the relative humidity (RH) and temperature. At room temperature and 1% RH, dehydration is driven by the diffusion mechanism, while under high RH (>30%), kinetic laws are typical of nucleation and growth mechanism. By increasing the RH, various metastability driven crystalline forms were obtained mimicking successive intermediate states between hydrate form and anhydrous form achieved under high RH. In contrast, the dehydration kinetics of caffeine hydrate under various RH levels can be described by only one master curve corresponding to a nucleation mechanism. Various metastability driven states were achieved depending on the RH, which can be described as intermediate between forms I and II of anhydrous caffeine.

## 1. Introduction

Organic molecular materials are characterized by a pronounced contrast between strong covalent intramolecular interactions and soft intermolecular attractions, including van der Waals potentials or hydrogen bonding association [1]. Slight temperature changes induce competition between kinetics and potential energies, responsible for specific physical properties of molecular compounds (rich polymorphism, molecular disorder, low melting temperature, etc.), while covalent bonds maintain the cohesion within the molecule. Interestingly, Raman spectroscopy provides the opportunity to analyze the three different types of molecular motions that exist in the framework of the rigid body model [1,2]:(i)The external motions corresponding to collective vibrations detected below 150 cm^−1^. In crystals, they are named lattice vibrations or phonons and they provide the crystalline fingerprint. Molecular disorder induces the broadening of phonon peaks up to the total disorder of the amorphous state, the low-frequency spectrum consisting of a vibrational density of states reflecting the short-range order. Information about collective motions can also be obtained from terahertz spectroscopy, with different selection rules inherent to the crystalline symmetry.(ii)The internal motions, corresponding to vibration within the molecule, including the 500–1800 cm^−1^ region distinctive of the molecular fingerprint and the X–H stretching region (between 2800 and 3800 cm^−1^, X = C, N, O), provide information about H-bonding associations.(iii)The semi-internal (or external) motions corresponding to very low-frequency rotations of a group of atoms within the molecule or the whole molecule.

A wealth of information contained in the low-frequency spectrum (LFRS) of molecular materials arises from the overlapping of external and semi-external motions. However, the processing of raw spectra is crucial and not trivial for extracting correct information about these two classes of motions. Additionally, the correct processing of raw LFRS requires the detection of the Raman signal at the very low frequencies. These two specific features (experimental spectrometer configuration and data processing) make the complete and correct extraction of information from LFRS very difficult. The development of a new generation of filters makes it possible to acquire the Raman signal at a low frequency with routine spectrometers [3]. Access of the low-frequency region provides very rapid direct information about the identification of crystalline polymorphism, contributing to the development of low-frequency Raman investigations in the field of pharmaceutical sciences [4,5,6,7,8,9,10,11,12,13]. In this context, the dissemination of information about spectra processing is of primary importance.

This paper first aimed to show the various types of information that can be extracted from LFRS and to describe the rigorous processing of raw spectra to obtain correct information. In the second step, LFRS was used to explore driven states obtained by dehydration theophylline and caffeine hydrates and to decipher the associated dehydration mechanisms. Previous investigations on caffeine have revealed metastability driven states only existing under milling [14], while other states are obtained by dehydration of caffeine hydrate without control of the relative humidity (RH) [15]. This study reports the comparison of the dehydration kinetics of caffeine and theophylline hydrates under various conditions of temperature and humidity. This study shows the high performances of low-frequency Raman spectroscopy for accurately determining the kinetic law of dehydration kinetics and exploring new metastable states achieved under stress.

## 2. Materials and Methods

### 2.1. Materials

Anhydrous theophylline (C_7_H_8_N_4_O_2_, purity ≥ 99%) was purchased from Sigma-Aldrich. Caffeine (purity = 98.5%) was purchased from Acros Organics. The hydrate forms of theophylline and caffeine were prepared by exposing the anhydrous forms to 98% RH for 24 h.

### 2.2. Methods

#### Raman Spectroscopy

Low-frequency Raman investigations were performed on a very high-dispersive XY-Dilor spectrometer, composed of three gratings in a configuration characterized by a focal length of 800 mm and equipped with a Cobolt laser emitting at 660 nm (250 mW). The monochromatic laser beam was focused on the powder sample via an achromatic lens for analyzing the largest possible volume of material (approximately 1 mm^3^). This configuration makes possible high elastic scattering rejection and a reduction in the detection of a Raman signal down to 5 cm^−1^. The acquisition time of each spectrum was 3 min, and they were collected in situ during dehydration within two different devices described below depending on the relative humidity and temperature conditions.Dehydration methods

Dehydration kinetics were carried out using the optical cell GenRH-M cell coupled to the RH generator GenRH-A purchased from Surface Measurement Systems Ltd. In situ hydration kinetics were performed at various RH and temperature levels. The temperature of the sample within the GenRH-M cell was controlled by the circulation of temperature-regulated water. Fluid temperature regulation between 5 and 60 °C is achieved using a F32-MC circulation cryostat Julabo. In the present study, the GenRH-M cell was regulated between 20 and 40 °C.

Dehydration kinetics were also analyzed without RH control using a THMS 600 Linkam temperature device. This temperature device was also used for analyzing anhydrous forms upon heating at 1 °C/min.

Data pre-processing

The raw Raman intensity *I*(*ω*,*T*) is considered as proportional to [16,17]:(1)$${\left(\omega_{0}-\omega \right)}^{4}\cdot \left(n\left(\omega \right)+1\right)\cdot S$$
where *ω*_0_ is the absolute frequency of the laser excitation in wavenumber units, ω is the Raman shift, (*n*(*ω*) + 1) is the Bose factor, and *S* is the intrinsic molar scattering activity. The proportionality of *I*(*ω*,*T*) to Expression (1) arises from the spectrometer response and the transparency of the material dependent on the laser radiation and its physical state. The ${\left(\omega_{0}-\omega \right)}^{4}$ dependence of *I*(*ω*,*T*) clearly shows the high sensitivity of LFRS, making rapid spectrum acquisition possible. As a consequence, low-frequency Raman spectra can be collected between 5 and 200 cm^−1^ in 1 min, in situ during heating ramp at 1 °C/min.

Equation (1) shows that the Raman intensity is directly proportional to the molar scattering intensity corresponding to the square of the derivative of the polarizability tensor with respect to the normal coordinate. The polarizability change generated by molecular vibration induces Raman activity.

The Raman band-shape in the low-frequency region is very sensitive to thermal fluctuations through the Bose factor and requires specific data processing to avoid the distortion of broad Raman bands in the low-frequency region. To resolve this issue, the Raman intensity ($I\left(\omega ,T\right)$) is transformed into reduced intensity ($I_{r}\left(\omega \right)$) according to [17,18]:(2)$$I_{r}\left(\omega \right)=\frac{I\left(\omega ,T\right)}{\left[n\left(\omega ,T\right)+1\right]\cdot \omega }$$

Representations of the LFRS of disordered molecular materials

The low-frequency spectrum of racemic ibuprofen (IBP) in the liquid state at 95 °C is represented at reduced intensity in Figure 1a. The spectrum is dominated by a very intense component, named quasi-elastic scattering (QES), in the very low-frequency range (<50 cm^−1^). This component reflects the thermal activation of rapid local motions, or mono molecular motions in very disordered molecular systems such liquids [19] or plastic crystals (rotator phases) [20,21]. As a consequence of its physical origin, this low-frequency contribution is highly temperature-dependent, as observed in Figure 1b. Thermal activation of these local motions, identified as corresponding to β-fast relaxational motions, is considered as directly involved in phase transition mechanisms. In this context, the analysis of the temperature dependence of the QES intensity (I_QES_) is an important issue for understanding phase transition mechanisms, requiring separation of the QES and pure vibrational component via the fitting procedure performed using Peakfit software v4.12, as described in Figure 1a. The contribution of QES is generally well described by a Lorentzian shape entered at ω = 0, while the contribution of vibrations in very disordered states is described by a lognormal shape. The lognormal shape reflects the distribution of various molecular packings forming cages, responsible for the inhomogeneous broadening of collective modes. This vibrational density of states (VDOS; *G*(*ω*)) is generally considered as well represented by Raman susceptibility via the conversion of the reduced intensity as [18,22]:(3)$$\chi \prime \prime \left(\omega \right)=\frac{C\left(\omega \right)}{\omega }G\left(\omega \right)=\omega I_{r}\left(\omega \right)$$
where *C*(*ω*) is the light vibration coupling coefficient.

The weak temperature dependence of the VDOS is typical of quasi-harmonic motions, as shown in Figure 2a. The Raman susceptibility representation is very sensitive to molecular disorder and makes it possible to compare disordered and ordered states, as reported in Figure 2b. Raman susceptibility of metastable phase II of IBP clearly reveals the substantial disorder of crystalline phase II, not detected in the structural resolution procedure from powder X-ray data. Figure 2b shows that the χ′′(ω) spectrum of the undercooled liquid state of IBP roughly corresponds to the envelope of lattice modes (phonon peaks) in the stable crystal. This is distinctive of the molecular organization of amorphous materials (glassy and liquid states), as shown in Figure 2a. Interestingly, the spectrum of phase II, typical of an amorphous state with the absence of phonon peaks, clearly exhibits shoulders associated with the phonon peaks of phase I. Consequently, the χ′′(ω) representation allows to describe the mechanism of the devitrification process of IBP, as a two-step transformation via a very disordered state and a metastable state composed of clusters of phase I.

Using LFRS to analyze the phase transformations in disordered molecular materials

Analysis of the temperature dependence of LFRS combining the analyses of QES and the χ′′(ω) spectrum provided a clear description of the devitrification mechanism of IBP. The temperature dependences of LFRS (at a reduced intensity) and I_QES_ are plotted in Figure 3 and compared with the DSC trace obtained under the same conditions as in the Raman investigations. Figure 3 shows the various transformations of the physical states of IBP. It is worth noting that the crystallization and melting of form II (FII) are clearly distinguishable, while both phenomena are overlapping in a complex DSC trace not easily interpretable. The plot of I_QES_(T) in Figure 3c also reveals the substantial disorder in FII compared with FI via the significant I_QES_ between the two phases.

## 3. Results

### 3.1. Analysis of the LFRS of Theophylline and Caffeine

The chemical structure of theophylline (TP) and caffeine (CAF) are plotted in Figure A1 in Appendix A. Despite the similarity of the chemical structures of both xanthine molecules, the LFRS of marketed materials, i.e., form II of TP and form II of CAF (plotted in Figure 4a), are significantly different. However, Figure 4a reveals that below 50 cm^−1^, the spectrum of CAF is the envelope of the low-frequency phonon peaks of TP. The relationship between these two commercial powders can be understood from the consideration of the orientational disorder of form I subsisting in form II of CAF. The Gibbs diagram of anhydrous CAF plotted in Figure A2 in Appendix B, representing an enantiotropic system [23], shows that form II transforms into the rotator phase I upon heating above 153 °C [24]. In form I, caffeine molecules slowly rotate around their six-fold molecular axis [25]. As a consequence, there is no three-dimensional periodicity of atomic positions; the Bragg peaks only result from the periodicity of the molecule mass centers [25]. The LFRS of rotator phases is dominated by the libration of molecules [20], as observed in plastic crystals. Form I of CAF is metastable at room temperature [26], making possible a comparison at room temperature of the LFRS of the two polymorphic forms in Raman susceptibility (Figure 4b).

The LFRS of form I reflects the librational density of states without detection of phonon peaks. The low-frequency region is the only spectral domain that allows clear identification of the two polymorphic forms. Indeed, the internal mode regions of the two phases being similar [20]. This similarity of the internal mode regions in the two phases indicates that the dynamical molecular disorder of form I is reminiscent of that of form II. This agrees with the atypical band-shape of the LFRS of crystalline form II composed of two broad bands, corresponding to the envelope of the low-frequency lattice modes of TP. The LFRS of form II of CAF can be naturally considered as reflecting the same kind of dynamical disorder as observed in form I. The splitting of the spectrum in form I into two components was interpreted as a consequence of the tilt of molecules out of the hexagonal plane of the lattice of crystalline form I, in a similar organization as determined in TP [27]. It is worth noting that only low-frequency Raman spectroscopy clearly revealed the existence of a dynamical disorder, while the structural resolution of form II from X-ray data was performed without consideration of disorder, leading to the determination of an unusually large unit cell ($V\sim 4277\;{Å}^{3}$) [28].

The hydrate forms of CAF (CAFh) and TP (TPh) were compared to anhydrates in Figure 5a,b, respectively. The comparison between hydrates (CAFh and TPh) is presented in Figure 5c. It is clear that the band located around 80 cm^−1^ in the spectrum of CAFh, which does not exist in the spectrum of forms I and II of CAF, is related to vibrations involving water molecules [15]. Figure 5c shows that this band also exists in the spectrum of TPh, rigorously superimposed with that in CAFh. Collecting the low-frequency spectrum in situ during the dehydration kinetics provided the opportunity to simultaneously analyze the lattice transformation and water escape.

#### 3.1.1. Analysis of TPh Dehydration

In the first step, only temperature was controlled in a Linkam temperature device. The determination of the transformation rate (ρ) is described in Appendix C. The dehydration kinetics were first analyzed without RH control. The time dependence of the transformation rate of TPh into anhydrous TP is plotted in Figure 6 at various temperatures.

The ρ(t) plot with a logarithm scale (Figure 6a) allows to compare the time of transformation, while the ρ(t) plot against t/t_1/2_ (Figure 6b), where t_1/2_ is the time to half-transformation, allows to compare the kinetic laws. The dehydration kinetics performed between 30 and 60 °C covered a wide time range from approximately ten to a thousand minutes. However, Figure 6b indicates that the kinetic laws followed a consistent behavior, except for dehydration at 60 °C, which showed some small deviations from this behavior. The time behavior had a clear sigmoidal shape associated with an Avrami-like function, $f\left(t\right)=1-exp\left(-k\cdot t^{n}\right)$. The fitting procedure led to the determination of n = 2.95 ± 0.02. This result can be interpreted as reflecting a nucleation and two-dimensional growth process of the anhydrous form. The low-frequency Raman spectra collected at the end of the dehydration kinetics were compared to the marketed form of TP in Figure 7.

Figure 7 reveals systematic spectrum changes between the dehydrated forms and marketed TP. These changes are highlighted by two arrows in Figure 7, which indicate an intensity increase in the very low-frequency region and an additional phonon peak around 25 cm^−1^ in spectra collected at the end of the dehydration kinetics. The intensity increase can be considered as corresponding to the quasi-elastic scattering, reflecting disordering induced by the water escape. On the contrary, the detection of an additional lattice mode in the dehydrated forms with respect to the marketed form reveals that the symmetry of the dehydrated forms was lower than that of the marketed TP and should be considered as reflecting a more ordered structural organization that that of the marketed TP. It is worth noting that all low-frequency spectra corresponding to anhydrates obtained at various temperatures exhibited similar band-shapes with the same number of phonon peaks. This indicates that dehydration at different temperatures leads to the same anhydrous form, although the additional band around 25 cm^−1^ was not clearly observed because of the broadening of phonon peaks induced by an increase in temperature. The fitting procedure of the low-frequency spectrum of the marketed and anhydrous forms (obtained by dehydration at 70 °C) is compared in Figure A6 in Appendix D. This figure clearly shows the spectral changes between the two anhydrous forms, including the existence of a 25 cm^−1^ band in the anhydrate obtained by dehydration at 70 °C.

In the second step, two series of dehydration were conducted by controlling the relative humidity using GenRH-M cell. Dehydration kinetics were first performed by maintaining 1% RH at T = 23, 30, 35, and 40 °C. The transformation rate calculated during these dehydration kinetics is plotted against the half-transformation time in Figure 8a. Additional dehydration kinetics were performed at various RH levels by maintaining the temperature at 35 °C. The corresponding dehydration kinetic curves are plotted in Figure 8b. In contrast to the dehydration kinetics performed without RH control, each kinetic law was different.

Time to half-transformation is plotted in Figure 9 for the various dehydration procedures performed by controlling the RH. It can be observed in Figure 8a (at a low RH) that the kinetic laws are similar in the early stages of dehydration and deviate gradually from that determined at 23 °C (indicated by dashed lines) with increasing temperature. In contrast to Figure 8a, Figure 8b shows different kinetic laws from the earliest stages of dehydration until the end. It can be observed that the kinetic law corresponding to the dehydration of TPh at 35 °C and 15% RH has a quasi-sigmoidal shape similar to that corresponding to the dehydration of TPh at 35 °C without RH control.

Spectra collected at the end of the dehydration kinetics performed at 35 °C were compared to the spectrum of the marketed form taken at 35 °C (Figure 10).

Figure 10 shows different band shapes in which the band distinctive of the presence of water molecules is absent. All of the kinetic laws plotted in Figure 8 indicate that the dehydration kinetics were completed, while the spectra plotted in Figure 10 suggest incomplete transformation, except for the spectrum taken after dehydration with 15% RH at 35 °C resembling the spectrum obtained by dehydration at 35 °C without RH control. All spectra collected at 35 °C and various RH levels correspond to metastable anhydrates, except those prepared at 15% RH and above. The kinetic law plotted at 1% RH was tentatively fitted using various identified models for describing the mechanism of solid-state transformation [29,30]. The best agreement was obtained with the following Jander equation [31]:(4)$${\left(1-{\left(1-\rho \right)}^{1/3}\right)}^{2}=kt$$
which is related to a three-dimensional diffusion mechanism. The fitting of the kinetic law is described in Figure 11. Above 1% RH, the kinetic laws deviated from the Jander equation earlier and earlier as the temperature increased. Intriguingly, Figure 10 shows that the longest dehydration, performed at 1% RH at 35 °C, led to the metastable state furthest from the stable anhydrate (obtained at 15% RH).

#### 3.1.2. Analysis of CAFh Dehydration

CAFh is recognized as not rigorously monohydrate but 4/5 hydrate, classified as a non-stochiometric hydrate [32]. The dehydration mechanism of CAFh was previously investigated by low-frequency Raman spectroscopy from kinetics performed at various temperatures without RH control [15]. In this paper, we report investigations of dehydration performed at various RH levels and room temperature (23 °C). The rate of solid-state transformation by dehydration was calculated from the method described in Figure A7 in Appendix E and plotted against time (t) and time to half-transformation (t/t_1/2_) in Figure 12a,b.

Figure 12a shows that the time of the solid-state transformation drastically increased above 10% RH. However, kinetic laws have a similar shape, indicating a single dehydration mechanism. Data collected during dehydration of CAFh at 20% RH and room temperature (23 °C) were carefully analyzed. The fitting of solid-state transformation is plotted in Figure 13 and compared with the water escape.

The best agreement between the experimental data and fitting curve was obtained with an Avrami-like model with exponent n = 1.410 ± 0.004. This value being close to 1 reflects a solid-state transformation driven by a nucleation mechanism with a quasi-absence of growth. Such a phenomenon was previously observed for the isothermal solid-state transformation of form I into form II at 90 °C [20]. Figure 13 shows that the solid-state transformation was closely related to the water escape.

The low-frequency spectra collected after dehydration at 5%, 10%, and 20% RH are plotted in Figure 14a. The three spectra were composed of two broad bands, as in form II. However, the band-shapes differed from each other and from that of form II. It can be observed that CAFh transformed into metastability driven states depending on the RH. These states can be considered as intermediate between forms I and II, but closer to form II. The separation of the two broad bands was more pronounced after dehydration at 20% RH. This indicates that CAFh transforms at room temperature into a metastability driven state increasingly close to form II as the RH increases. In contrast, Figure 14b shows that the spectra collected after dehydration under a high RH (>30%) increasingly resembled that of form I as the temperature of dehydration increased.

## 4. Discussion

Dehydration of theophylline and caffeine hydrates has attracted wide attention [15,32,33,34,35,36,37,38,39,40,41], particularly the identification of rich polymorphisms such as theophylline. It is recognized that theophylline presents a rich anhydrous polymorphism. Indeed, four anhydrates were identified, named differently in the literature, obtained via different drying routes. The marketed anhydrous form is named form II, in agreement with Seton et al. [37]. Form I can be obtained from evaporation at 100 °C of a saturated aqueous solution of TP. Form III is produced by dehydration of the hydrate form at low pressure [39,40,41]. Form IV is generated by slurrying form II in methanol [37].

The present study showed that dehydration at various temperatures without humidity control (estimated >30% RH) leads to a stable anhydrous form. Despite different band shapes of the low-frequency spectra plotted in Figure 7, distinctive of the dehydration temperature, all spectra corresponded to the same crystalline form, different from the marketed form. It was then considered as corresponding to form I. Dehydration performed at 35 °C and under increasing RH levels led to a panel of spectra increasingly resembling that obtained by dehydration of TPh at 35 °C without RH control. The kinetic laws plotted in Figure 8a indicate that the solid-state transformation was completed, while the various spectra show intermediate solid states between TPh and the anhydrous form I. The spectra obtained by dehydration under 1% RH at various temperatures are plotted in Figure 15a. The same gradual change of the low-frequency band-shape was observed as the temperature increased, reflecting different metastability driven states achieved by dehydration. In order to identify these metastable states, TPh was dehydrated at room temperature under primary vacuum (~ 5.10^−3^ mbar) for 12 h. The low-frequency spectrum of the anhydrous form was compared with that obtained by dehydration at room temperature under 1% RH (Figure 15b). Both spectra were almost superimposed, indicating that the anhydrous form obtained at 23 °C under 1% RH corresponds to form III. It is worth noting that the spectrum of form III plotted in Figure 15b rigorously corresponds to that reported in a recent paper [42]. Consequently, this study showed that numerous metastable states can be prepared upon dehydration depending on both temperature and RH. It was also shown that the mechanism of solid-state transformation is also dependent on RH and temperature.

The highly metastable character of form III was shown by heating the anhydrate obtained by dehydration under 1% RH at 23 °C. The spectra collected upon heating are plotted in Figure 16. As soon as the temperature increased, the spectrum drastically changed. From 40 °C, the spectrum already resembled the spectrum of form I.

At room temperature under very low RH, the transformation was mostly driven by a diffusion mechanism. The mechanism of solid-state transformation was modified by increasing the temperature, which promoted the nucleation and growth process.

Dehydration of CAFh at various RH levels also lead to various metastability driven states different from forms I and II, as previously observed without RH control [15]. In contrast to TPh, the mechanism of dehydration of CAFh followed the same master curve independently of the RH. This could be related to the high metastability of the CAFh inherent to the non-stochiometric character of hydrated caffeine. The possibility of achieving multiple metastable states under dehydration could have consequences on the drug solubility.

## 5. Conclusions

The present study revealed the existence of multiple states driven by dehydration of TPh and CAFh at various temperature and RH levels. It was shown that anhydrous form III of TP obtained by dehydration under low pressure can also be obtained at a very low RH (1%) at room temperature (23 °C). Only low-frequency Raman spectroscopy allowed to obtain such kinds of information in caffeine and theophylline. It has already observed that only low-frequency Raman spectroscopy makes it possible to explore metastability driven states of caffeine under milling [14] or metastable states induced by tableting [43]. This study suggests the possible existence of metastable states induced by tableting various theophylline forms, which could be revealed by low-frequency Raman investigations. More generally, the easy use of Raman spectroscopy requiring no specific sample preparation and the development of new filters providing routine access to the low-frequency region make low-frequency Raman spectroscopy a promising technique to rigorously control the physical state of APIs under various types of stress.

## Figures and Tables

**Figure 1 pharmaceutics-15-01955-f001:**
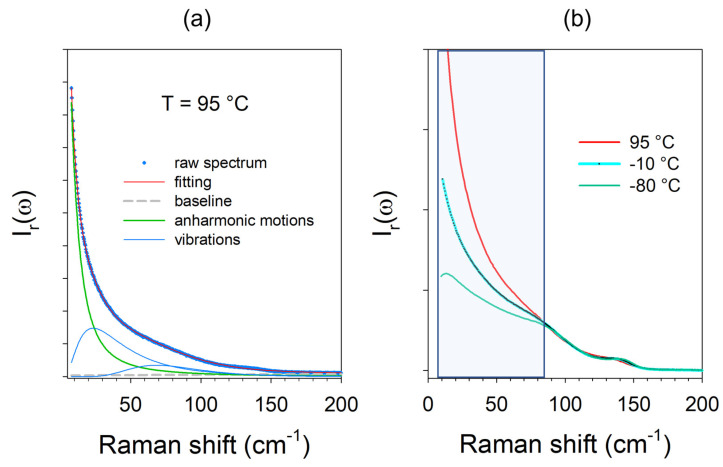
Low-frequency spectra of ibuprofen at a reduced intensity. (**a**) Description of the fitting procedure of the spectrum collected in the liquid state. (**b**) Temperature dependence of the spectrum collected in a glass (−80 °C) in undercooled liquid (−10 °C) and in liquid at 95 °C. The colored region highlights the contribution of the quasi-elastic scattering.

**Figure 2 pharmaceutics-15-01955-f002:**
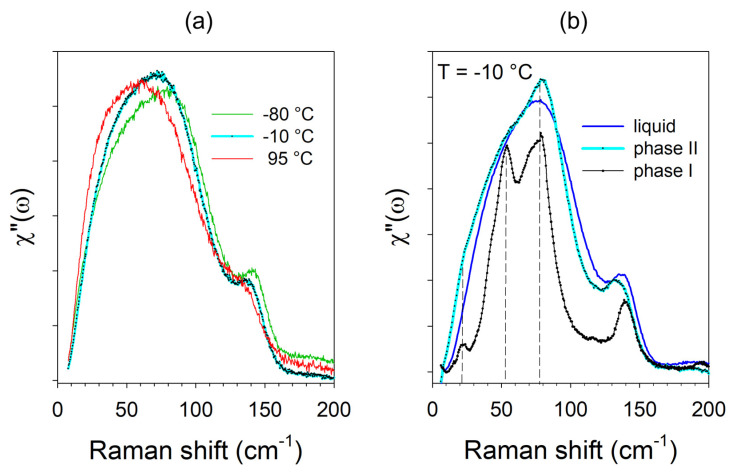
Low-frequency spectra of ibuprofen plotted in Raman susceptibility. (**a**) Temperature dependence of the spectrum plotted in various amorphous states. (**b**) The spectra of the crystalline forms of ibuprofen compared with that of the undercooled liquid taken at the same temperature (−10 °C).

**Figure 3 pharmaceutics-15-01955-f003:**
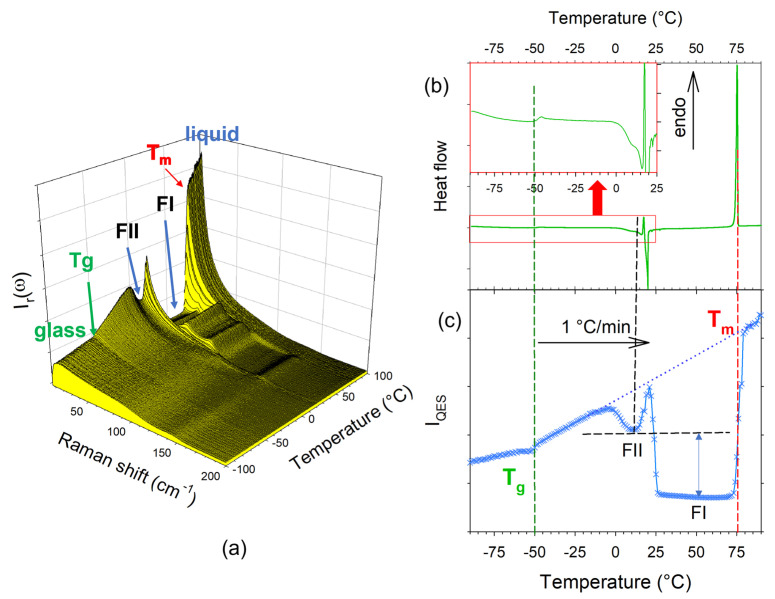
Analysis of the phase transition sequence of ibuprofen upon heating at 1 °C/min from the glassy to liquid states. (**a**) Temperature dependence of the spectrum plotted at a reduced intensity. (**b**) DSC trace taken with the same scanning rate. (**c**) Comparison with the temperature dependence of the quasi-elastic intensity. The double arrow highlights the disorder in form II compared with form I; vertical dashed lines show the correspondence between the signatures of the phase transitions distinctive of the DSC and LFRS techniques.

**Figure 4 pharmaceutics-15-01955-f004:**
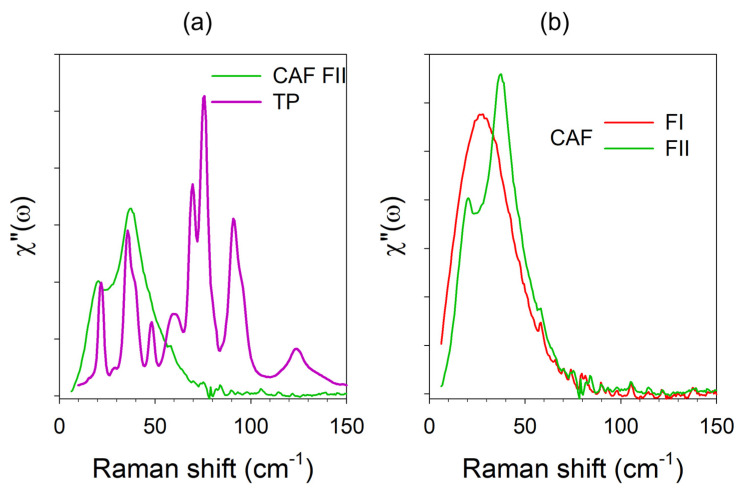
Low-frequency Raman spectra of theophylline and caffeine plotted in Raman susceptibility at room temperature. (**a**) Marketed crystalline forms. (**b**) Disordered forms I and II of caffeine at room temperature.

**Figure 5 pharmaceutics-15-01955-f005:**
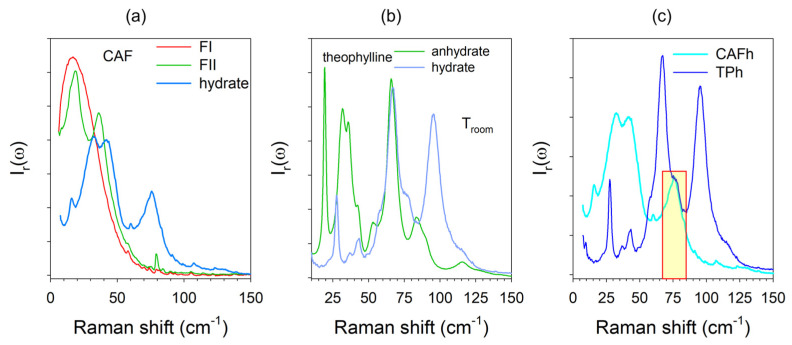
Low-frequency Raman spectra of the hydrate and anhydrate forms of caffeine and theophylline plotted at a reduced intensity. (**a**) Hydrate compared with the anhydrous forms I and II of caffeine. (**b**) Hydrate and marketed anhydrate forms of theophylline. (**c**) Comparison of the hydrate forms of caffeine and theophylline. The region colored in yellow highlights the Raman bands involving vibrations of water molecules.

**Figure 6 pharmaceutics-15-01955-f006:**
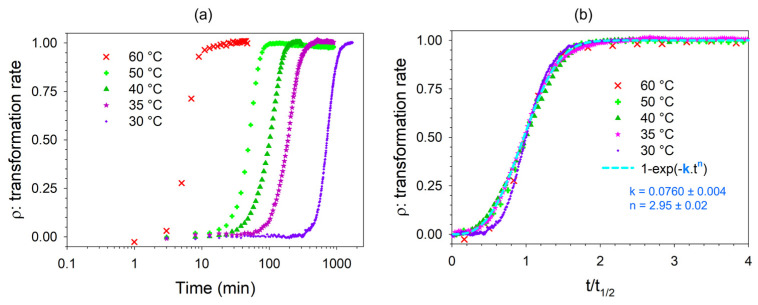
The rate of solid-state transformation of theophylline during dehydration at various temperatures plotted against (**a**) time and (**b**) the ratio of time over time to half-transformation.

**Figure 7 pharmaceutics-15-01955-f007:**
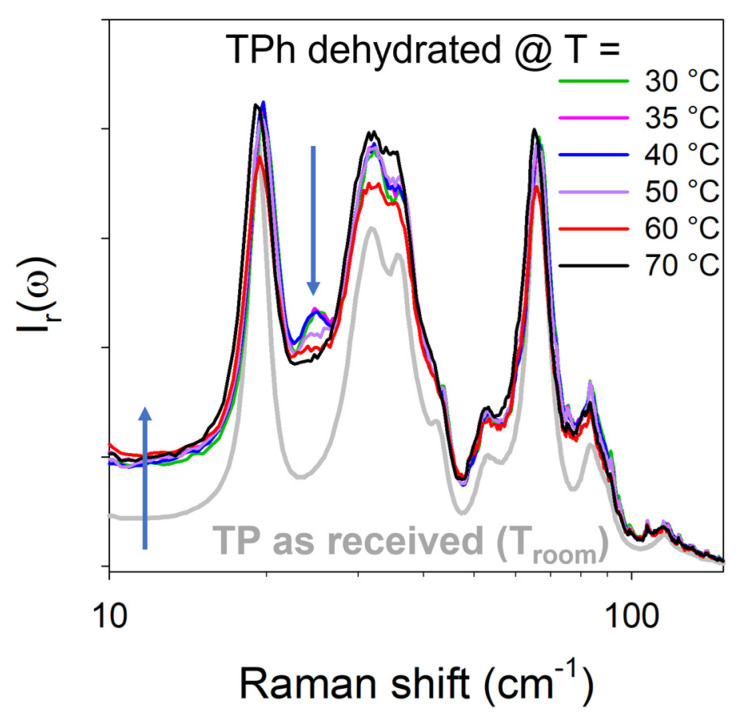
Low-frequency spectra of theophylline collected after dehydration at various temperatures compared with that of the marketed form (TP as received). The arrows indicate the two major spectral modifications between the marketed and dehydrated forms. Logarithmic ω-scale was used for better clarity in the low-frequency region.

**Figure 8 pharmaceutics-15-01955-f008:**
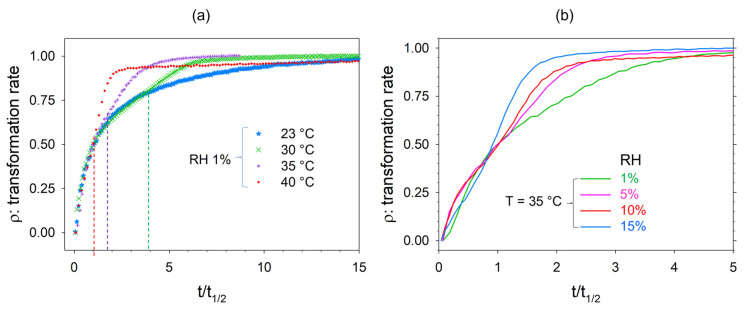
The rate of the solid-state transformation of theophylline during dehydration (**a**) under 1% RH at various temperatures, where dashed lines indicate deviations of the kinetic law determined at 30, 35, and 40 °C from that determined at 23 °C; (**b**) at 35 °C under various RH levels.

**Figure 9 pharmaceutics-15-01955-f009:**
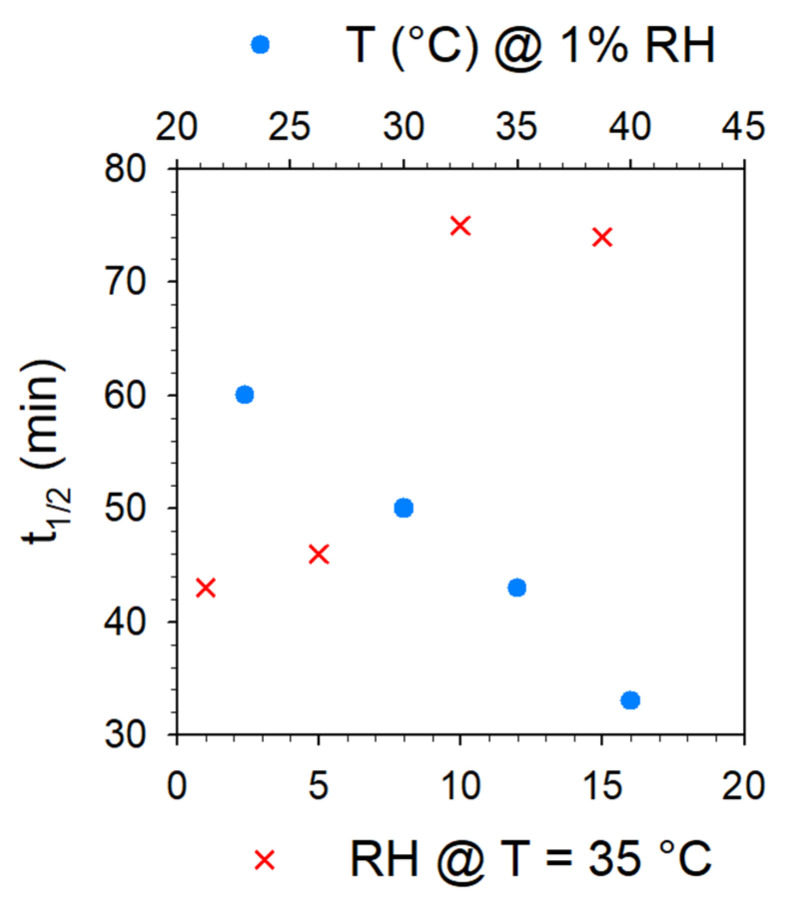
Plot of time to half-transformation plotted for dehydration under various RH levels at 35 °C and at various temperatures under 1% RH.

**Figure 10 pharmaceutics-15-01955-f010:**
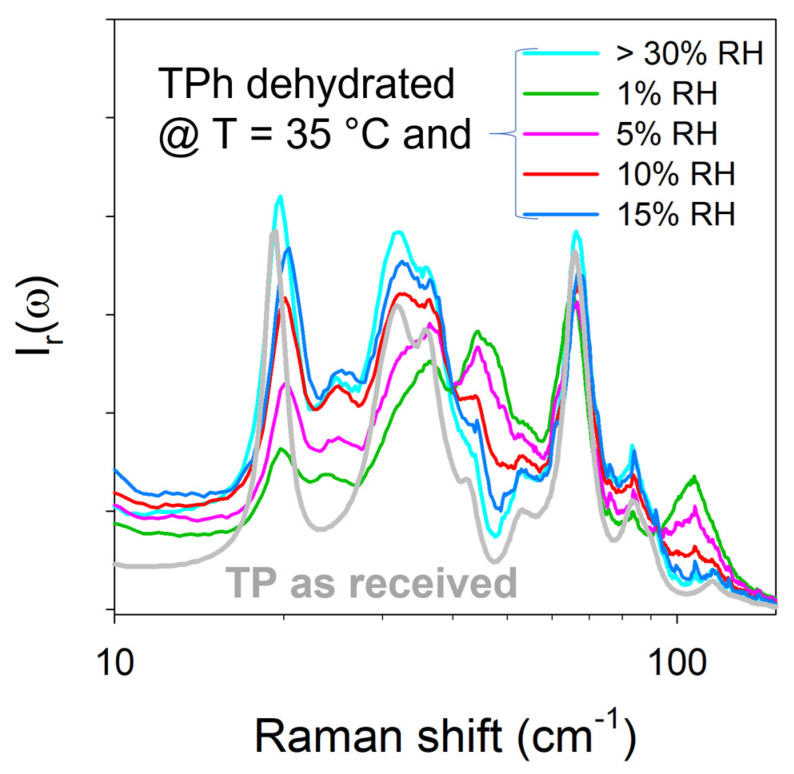
Low-frequency spectra of theophylline collected after dehydration at 35 °C under various RH levels compared with those of the marketed form (TP as received) and the anhydrate obtained at 35 °C without RH control (estimated >30%). Logarithmic ω-scale was used for better clarity in the low-frequency region.

**Figure 11 pharmaceutics-15-01955-f011:**
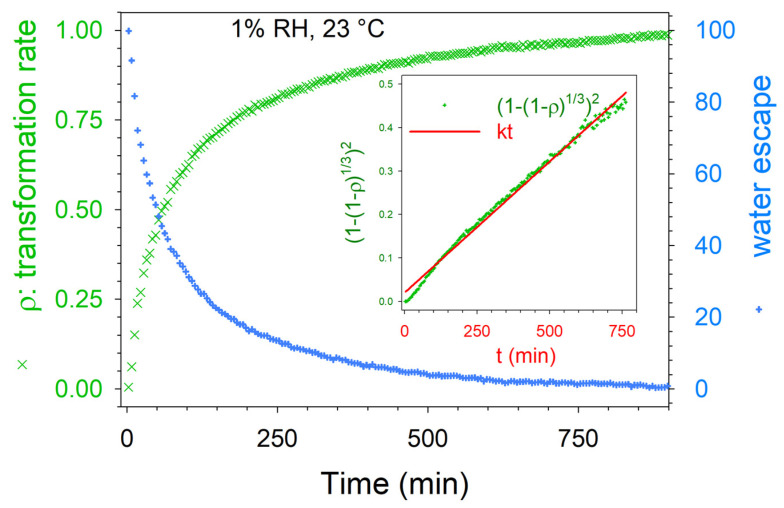
Solid-state transformation rate and water escape calculated during dehydration at 23 °C under 1% RH plotted against time. The fitting procedure of ρ(t) with Equation (4) is plotted in the inset.

**Figure 12 pharmaceutics-15-01955-f012:**
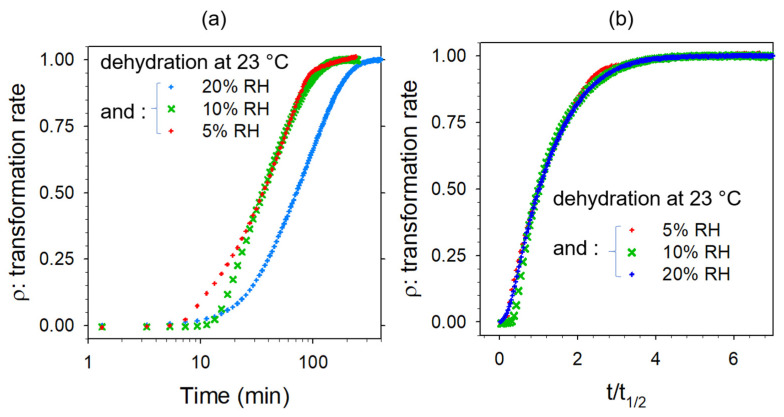
The rate of the solid-state transformation of caffeine during dehydration at 23 °C under various RH levels plotted against (**a**) time and (**b**) the ratio of time over time to half-transformation.

**Figure 13 pharmaceutics-15-01955-f013:**
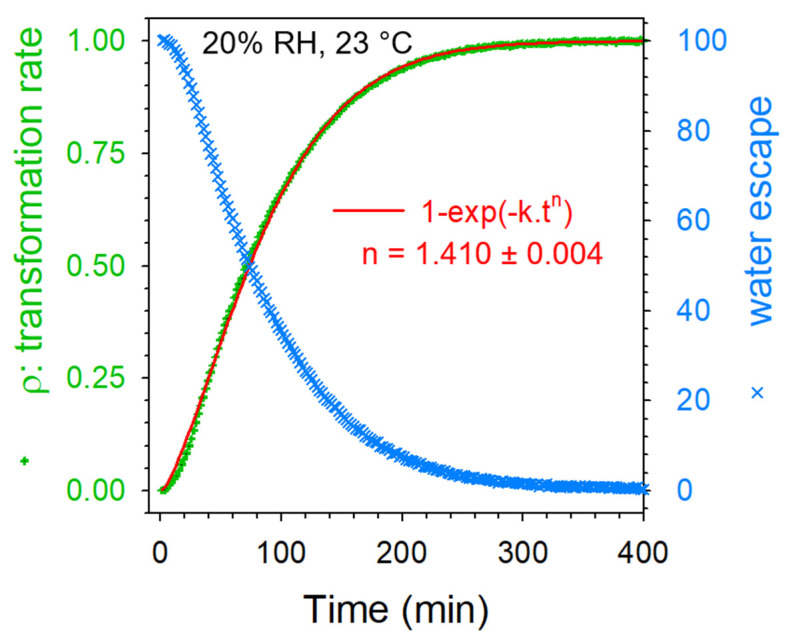
Rate of solid-state transformation and water escape calculated during dehydration of caffeine at 23 °C under 20% RH plotted against time.

**Figure 14 pharmaceutics-15-01955-f014:**
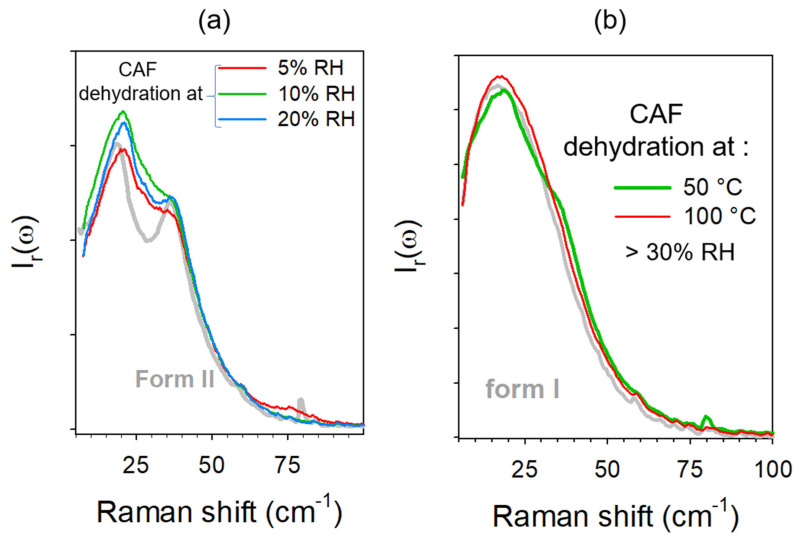
Low-frequency spectra of anhydrates obtained by dehydration of caffeine hydrate (**a**) at 23 °C under various RH and (**b**) at various temperatures without RH (>30%). Spectra obtained after dehydration were compared with the spectra of forms II (**a**) and I (**b**).

**Figure 15 pharmaceutics-15-01955-f015:**
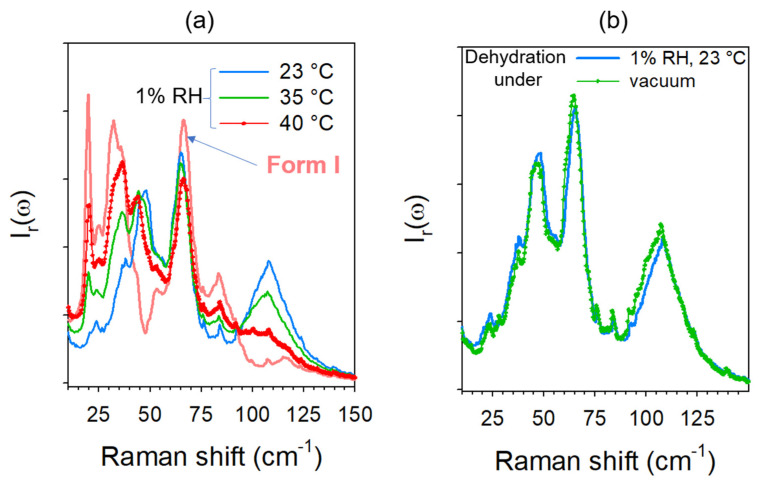
Low-frequency spectra of theophylline anhydrates. (**a**) Anhydrates prepared under 1% at various temperatures compared with form I. (**b**) Comparison between anhydrates prepared at room temperature under 1% RH and low pressure.

**Figure 16 pharmaceutics-15-01955-f016:**
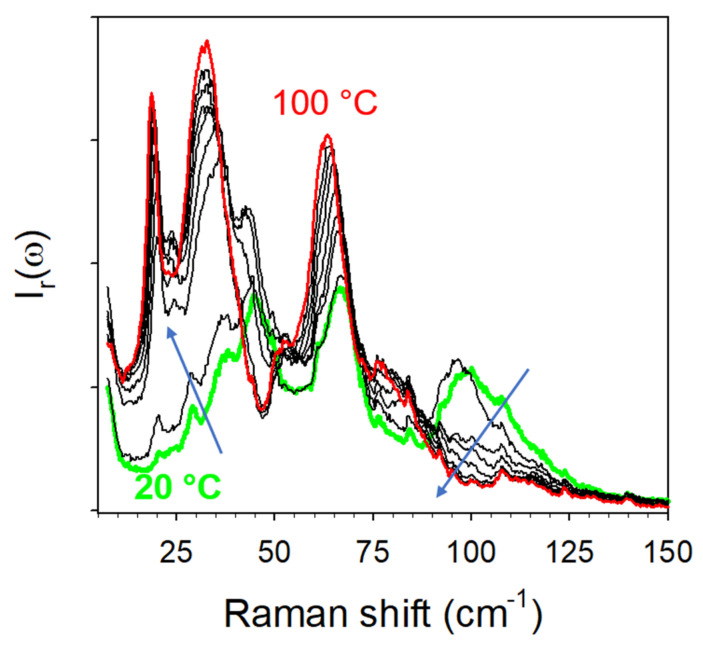
Low-frequency spectra collected upon heating form III at 1 °C/min, plotted by steps of 10 °C. The arrows highlight the most significant changes in the spectrum.

## Data Availability

Data are shown within the article.

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
