# Peer review of "Low-Frequency Raman Spectroscopy: An Exceptional Tool for Exploring Metastability Driven States Induced by Dehydration"

_pharmaceutics, 2023, doi:10.3390/pharmaceutics15071955_

Round 1

Reviewer 1 Report

The use of low-frequency Raman spectroscopy (LFRS) is increasing in importance in the field pharmaceuticals. LFRS has been used to examine the dehydration mechanism of two hydrates (theophylline and caffeine)  In the cse of theophylline, two distinct mechanisms of solid-state transformation were identified  These depended on the relative humidity (RH) and temperature. At room temperature and 1% RH, dehydration occurs via a diffusion mechanism, while at high RH (>30%), the kinetics follow nucleation and growth mechanisms. With increasing RH, a range of metastable crystalline forms  between the hydrate form and the anhydrous form could be detected. LFRS demonstrates that the anhydrous form III of theophylline, obtained through low-pressure dehydration, can also be achieved at very low RH (1%) and room temperature (23 °C). In contrast, the dehydration kinetics of caffeine hydrate under different RH conditions were described by a nucleation mechanism. LFRS data for caffeine at different  RH levels appears to show metastable intermediates that lie spectrally between forms I and II of anhydrous caffeine.

The paper does a very nice job of explaining the use of reduced intensity in presenting Raman spectra; this has important consequences for the effectiveness of interpretation.  The work builds on existing studies in a meaningful, purposeful and significant way.

Author Response

no correction requested

Reviewer 2 Report

The entitled manuscript “Low-frequency Raman spectroscopy: an exceptional tool for exploring metastable driven states induced by dehydration” by Elisabeth Fink et al intend to analyze dehydration mechanism of two hydrates using LFRS. In my opinion, the manuscript seems to be suitable for publication in the journal of pharmaceutics. 

Author Response

no correction requested

Reviewer 3 Report

This manuscript deals with the hydration and dehydration mechanisms of theophylline and caffeine, presenting the usefulness of the low-frequency Raman spectroscopic measurements. The experiments were conducted in different well-defined conditions using various temperatures and humidities. The analyses of spectra acquired by the time are comprehensive, and the conclusions are correct. Only a minor revision is required before publication.

Comment:

Perhaps a comparison with the results of the study by Paiva et al. (DOI: 10.1021/acs.molpharmaceut.1c00476) should be incorporated into the manuscript. The Discussion part (Section 4) may be an appropriate part of the manuscript to do that.

Reviewer 4 Report

Interesting work, certainly worthy of publication. The paper illustrates the possibilities of low-frequency Raman spectroscopy, and also publishes new scientific data. However, this led to an increase in the volume of the article, which somewhat complicates its assessment and perception.

The text of the work is overloaded with figures. Moreover, sometimes the data in the figures are duplicated (see, for example, figures 4 and 5). Perhaps it pays to exclude some figures from the text, or may be present them as tables (this is an optional requirement).

Some remarks:

1. What is the radiation power (at the object) during measurements?

2. What software was used for plotting and analysis?

3. Perhaps these are problems of my text editor, but formulas 1 and 2 are not displayed quite correctly in the text (incomprehensible punctuation points between symbols). It is necessary to correct formulas 1 and 2 for correct display in the text (???)?. ?. (?(?) + ?). ?

4. Line 247 - figure 6c is mentioned. Where is this figure in the text?

5. Complete the caption of figure 11. It is not clear what this is about.

Reviewer 5 Report

This is a very comprehensive description and demonstration of the applications of Low Frequency Raman Spectroscopy to analysis of pharmacological compounds. It is well conceived and presented, although there are some minor issues which should be considered to improve the quality of presentation;

(i) It would be useful in he Abstract and point number (iii)  in the introduction to specify what is considered LFRS (e.g. < xxx cm-1)

(ii) The authors should also comment, in the Introduction, on the relationship with  what is called THz spectroscopy

(iii) "the bose factor" - should be capitalised - please also provide a reference

(iv) "to overcome the distortion of broad Raman bands in the low-frequency region", please elaborate to better explain this, or provide reference

(v) "However, the spectrum of CAF is the envelope of the low-frequency phonon peaks of TP." The meaning of this statement is also not very clear

(vi) in some places, the language is somewhat imprecise - for example, should this mean  "The relationship between THE RAMAN SPECTRA OF these two commercial powders.."

(vii) In Figure 6. - the y-axis title should have units

(viii) "However, Figure 6c indicates..." Figure 6b, I think

(ix) "the kinetic laws follow a master curve, except for dehydration at 60 °C which shows some small deviations from the master curve"  - "a consistent behaviour" is better than a "master curve". In fact, the 60 °C curve only deviates in two points...

(x) In Figure 6/7 - "marketed form" is not evident in the figures, as it is called "as received" - these should be more consistent

(xi) "Jander equation" - should be numbered and referenced

(xii) The statements below should each be referenced:

CAFh is recognized as not rigorously monohydrate but 4/5 hydrate and then classified asa non-stochiometric hydrate.

Dehydration mechanism of CAFh was previously investigated by low-frequency Raman spectroscopy from kinetics performed at various  temperatures without RH control.

The English is generally good, but could do with a final proofing
